# Jasmonate and Ethylene-Regulated Ethylene Response Factor 22 Promotes Lanolin-Induced Anthocyanin Biosynthesis in ‘Zaosu’ Pear (*Pyrus bretschneideri* Rehd.) Fruit

**DOI:** 10.3390/biom10020278

**Published:** 2020-02-11

**Authors:** Ting Wu, Han-Ting Liu, Guang-Ping Zhao, Jun-Xing Song, Xiao-Li Wang, Cheng-Quan Yang, Rui Zhai, Zhi-Gang Wang, Feng-Wang Ma, Ling-Fei Xu

**Affiliations:** College of Horticulture, Northwest A&F University, Taicheng Road NO.3, Yangling 712100, Shaanxi Province, China; 18829351538@163.com (T.W.); hylht3598@163.com (H.-T.L.); zhaogp1996@nwafu.edu.cn (G.-P.Z.); JunxingSong@163.com (J.-X.S.); XiaoLiW365@163.com (X.-L.W.); cqyang@nwsuaf.edu.cn (C.-Q.Y.); zhongdishaonian@sina.com (R.Z.); fwm64@sina.com (F.-W.M.)

**Keywords:** pear, anthocyanin, *PbERF22*, MYB transcription factors, ethylene, jasmonate

## Abstract

Anthocyanin contributes to the coloration of pear fruit and enhances plant defenses. Members of the ethylene response factor (ERF) family play vital roles in hormone and stress signaling and are involved in anthocyanin biosynthesis. Here, *PbERF22* was identified from the lanolin-induced red fruit of ‘Zaosu’ pear (*Pyrus bretschneideri* Rehd.) using a comparative transcriptome analysis. Its expression level was up- and down-regulated by methyl jasmonate and 1-methylcyclopropene plus lanolin treatments, respectively, which indicated that *PbERF22* responded to the jasmonate- and ethylene-signaling pathways. In addition, transiently overexpressed *PbERF22* induced anthocyanin biosynthesis in ‘Zaosu’ fruit, and a quantitative PCR analysis further confirmed that *PbERF22* facilitated the expression of anthocyanin biosynthetic structural and regulatory genes. Moreover, a dual luciferase assay showed that *PbERF22* enhanced the activation effects of *PbMYB10* and *PbMYB10b* on the *PbUFGT* promoter. Therefore, *PbERF22* responses to jasmonate and ethylene signals and regulates anthocyanin biosynthesis. This provides a new perspective on the correlation between jasmonate–ethylene crosstalk and anthocyanin biosynthesis.

## 1. Introduction

Anthocyanins, an important class of plant secondary metabolites, play important roles in plant growth and development and in resistance to environmental stresses [1]. The color affects fruit quality, and the formation of fruit pigments depends on anthocyanin synthesis during fruit growth and development. Moreover, in plants, accumulated anthocyanins enhance resistance to biotic and abiotic stresses, such as UV-B light damage [2], pathogen infection [3], wounding [4], and drought [5].

The anthocyanin biosynthetic pathway has been researched and illuminated in many species. A series of enzymes encoded by structural genes (*PAL*, *CHS*, *CHI*, *F3H*, *DFR*, *ANS*, and *UFGT*) are involved in anthocyanin biosynthesis, and these structural genes are co-regulated by the MBW transcription complex composed of myeloblastosis (MYB), basic helix-loop-helix proteins (bHLH), and WD40 proteins. In pear, flavonoid 3-O-glucosyltranferase (UFGT) is the key enzyme that determines the anthocyanin content in the fruit peel [6,7]. *MYB10.1* isolated from pear *(Pyrus pyrifolia* cv. Aoguan) is similar to *MYB10*, and it interacts with bHLH3 to promote anthocyanin biosynthesis in pear [8]. In apple, *MdMYB1* is positively correlated with anthocyanin biosynthesis [9]. The overexpression of *MdMYB10* significantly enhances anthocyanin accumulation [10]. In *Prunus*, the overexpression of *MYB10.1*/*bHLH3* and *MYB10.3*/*bHLH3* increases the expression levels of *NtCHS*, *NtDFR,* and *NtUFGT* to promote anthocyanin synthesis [11]. In addition to the *MYB*-*bHLH*-*WD40* (MBW) complex, other transcription factors can coordinately regulate the expression of genes involved in anthocyanin synthesis. *Arabidopsis thaliana ZAT6* directly binds the promoters of *MYB12* and *MYB111* to activate the expression levels of genes involved in anthocyanin biosynthesis in plants treated with H_2_O_2_ [12]. MdWRKY40 interacts with MdMYB1 to promote wounding-induced anthocyanin biosynthesis in apple [13]. *PpHY5* directly binds to the promoter regions of *PpCHS*, *PpDFR*, *PpANS*, and *PpMYB10*, thereby promoting anthocyanin accumulation [14]. 

The ethylene response factor (ERF) family, belonging to the AP2/ERF transcription factor family, plays an essential role in plant growth, and its members can be stimulated by ethylene, jasmonate, abscisic acid, and auxin. They are involved in regulating various processes in plants, such as defense and stress responses [15]. MdERF3 interacts with MdMYC2 to activate *MdACS1* transcription, thereby participating in jasmonate- and ethylene-mediated apple fruit ripening [16]. ERF1 responses to ethylene and jasmonic acid play important roles in ethylene/jasmonic acid-dependent defense responses [17]. The ERF proteins enhance plant defense systems by activating a pathogen-inducible plant defensing, PDF1.2 [18]. Moreover, *ERF* genes appear to regulate anthocyanin synthesis in various species. Under drought-stress conditions, the interaction between MdERF38 and MdMYB1 promotes anthocyanin biosynthesis [19]. *MdERF1B* regulates anthocyanin biosynthesis by interacting with MdMYB9 and MdMYB11 in apple [20]. The down-regulation of *JcERF035* accelerates anthocyanin accumulation under low-phosphate conditions in Arabidopsis [21]. *PyERF3* interacts with MYB114 and forms a new complex with bHLH3 to co-regulate anthocyanin biosynthesis [22]. *Pp4ERF24* and *Pp12ERF96* are involved in fruit coloration and the anthocyanin synthesis induced by blue light in pear [23].

Lanolin, a complex mixture, consisting of esters of sterols, triterpene alcohols, esters of aliphatic alcohols, and mono-hydroxyesters of sterols, triterpenes, and aliphatic alcohols, has wide applications not only in chemical and medicinal industries [24], but also in plant research. It has been used extensively in plant research as a carrier for plant growth regulators [25] and as a main component of fungicides [26]. The earliest research on the use of lanolin in plants was carried out by Laibech et al. [27,28]. It has since been used as a carrier for plant growth regulators in many studies on the physiological responses of plants, such as during parthenocarpy [29], fruit development [30], and the rooting of cuttings [31], with pure lanolin treatments serving as negative controls in these studies. Lanolin alone has no significant physiological effects on some plant tissues, such as roots [32], stems [33], leaves [34], and flowers [35]. However, lanolin can affect other plant tissues sometimes. Applications of lanolin paste influence periderm development and tissue morphology in the rinds of melon fruits [36], and lanolin alone promotes shoot initiation from the axillary buds on the laterals of pruned kiwifruit vines [37].

In this study, we found an interesting phenomenon: lanolin was used on the fruit skin to cause the red coloration of the ‘Zaosu’ pear fruit. We further investigated the mechanisms of the skin coloration in pear and the anthocyanin biosynthetic pathway induced by this substance. *PbERF22*, as the candidate gene, was screened from RNA sequencing (RNA-Seq) data. We investigated its relationships with the two hormones jasmonate and ethylene and further verified the role of *PbERF22* in anthocyanin biosynthesis. The overexpression of *PbERF22* in pear confirmed the mechanism by which *PbERF22* regulates anthocyanin accumulation, and a dual luciferase (LUC) assay revealed that *PbERF22* enhances *PbUFGT* promoter activity. *PbERF22* may regulate anthocyanin biosynthesis by enhancing the activation effects of *PbMYB10* and *PbMYB10b* on the *PbUFGT* promoter. These studies provide a new perspective on the potential mechanisms of anthocyanin biosynthesis in green-skinned pear.

## 2. Materials and Methods

### 2.1. Plant Materials and Treatments

The experiment was carried out on a plantation in Wugong, Shaanxi Province, China, in May 2018. In total, 30 ‘Zaosu’ fruit skins were coated with a thin layer of anhydrous lanolin (Xi’an Tianzheng Pharmaceutical Accessories Co. LTD’, Xian, China; CAS: 8006-54-0) at 60 days after full bloom (DAFB), and untreated ‘Zaosu’ fruit were used as controls. Treated and untreated fruits were harvested after 10 and 20 days of normal growth. They were immediately transported to the laboratory on the same day. The lanolin remaining on the peel was removed with 75% alcohol before the skin was peeled off. The peels of five fruit were treated as a biological replicate. The peels of the fruit were peeled off. All the tissues were analyzed using three biological replicates. Isolated fruit peels were immediately frozen in liquid nitrogen and then stored at −80 °C for further use.

The 1-methylcyclopropene (1-MCP; an ethylene antagonist; Rohm & Haas, Philadelphia, Pennsylvania, PH, USA) was applied to fruit using the method described by Freiman et al. [38]. In June 2019, the ‘Zaosu’ pear fruit from the plantation in Wugong, Shaanxi Province, China, were treated at 90 DAFB with 1-MCP. Each fruit was wrapped in a plastic bag (low-density polyethylene, 16 cm × 26 cm, 60-μm thick). Before sealing the bag, approximately 80 mg/L 1-MCP was released into the bag. The bag was removed after 24 h. Bagged untreated fruit served as the controls. Then, half of the untreated and 1-MCP-treated fruits were coated with lanolin, while the remaining half were not treated. After 10 days, all the fruit was harvested. Each treatment contained three biological replicates, each including five fruit.

A total of 100 ‘Zaosu’ fruit were obtained 100 DAFB from an orchard in Meixian, Shaanxi Province, China, in July 2019, and they were immediately transported to the laboratory. The collected fruit was divided into two groups; one group was treated with water, and the other was treated by soaking in 2 mmol/L methyl jasmonate (MeJA; Sigma-Aldrich, Sigma-Aldrich, St Louis, MO, USA) for 5 min at room temperature, as described by Ma et al. [39]. The two fruit groups were dried and then placed at room temperature for 8 days. They were sampled every 4 days. The peels of five fruit served as a biological replicate. The fruit peels were peeled off. All the tissues were analyzed using three biological replicates. Isolated fruit peels were immediately frozen in liquid nitrogen and then stored at −80 °C for further use.

### 2.2. Measurement of the Anthocyanin Contents

The total anthocyanin extraction was performed using the slightly modified method of Giusti and Wrolstad (2001) [40]. Briefly, the peel tissue was quickly ground into a powder in liquid nitrogen, 0.2 g was weighed, and 2 mL of the 1% HCL-methanol solution was added. Next, the specific extraction methods and anthocyanin content calculations were performed using the method of Wang et al. [41]. The absorbance levels of extracts at 520 nm and 700 nm were determined using a UV-Visible spectrophotometer (UV-1700, Kyoto, Japan). The total anthocyanin content is presented as mg/100 g fresh weight. The value for each sample is expressed as the means of three independent biological replicates.

### 2.3. RNA Extraction and cDNA Synthesis

Fruit from each sampling point were divided into three groups, with each group containing five fruit. The peels in each group were evenly mixed for the total RNA extraction and used as one biological replication. The total RNA extraction was carried out using an RNAprep Pure Plant Kit (TIANGEN, Beijing, China) in accordance with the manufacturer’s instructions. The RNA concentration and quality were tested using a UV-Visible spectrophotometer (UV-1700, Kyoto, Japan). The first-strand cDNA was synthesized from 1 µg of total RNA using a PrimeScript RT reagent kit with gDNA Eraser (TaKaRa, Dalian, China) and used for quantitative real-time PCR assays.

### 2.4. Library Construction and Transcriptome Analysis

Samples of lanolin-treated and untreated control ‘Zaosu’ fruits peel from 10 and 20 days after treatment were used to construct RNA-Seq libraries. The 12 RNA-Seq libraries constructed were as follows: Untreated-10 (10-day control), Untreated-20 (20-day control), Lanolin-10 (10 days after lanolin-treatment), and Lanolin-20 (20 days after lanolin-treatment), each with three biological replicates. All the libraries were obtained using the following steps: total RNA detection, mRNA enrichment, double-stranded cDNA synthesis, purified double-stranded cDNA’ terminal repair, AMPure XP bead-based fragment selection, PCR-amplification enrichment, and quality control analyses.

The libraries were used to carry out Illumina HiSeq sequencing. To ensure the quality of the RNA-sequencing information analysis, raw reads were filtered to select clean reads and eliminate low-quality reads with adapters. The clean reads obtained were used in the subsequent statistical analysis. The gene expression levels were estimated using the expected number of fragments per kilobase of transcript sequence per millions base pairs sequenced method. The genes that met the screening standards of padj < 0.05 and |log2 (ratio)| > 1 were defined as differentially expressed genes (DEGs). The pathway enrichment analyses of Kyoto Encyclopedia of Genes and Genomes (KEGG) were conducted using the website platform (www.genome.jp/kegg/).

### 2.5. Quantitative Real-Time PCR Validation

The qRT-PCR reactions contained SYBR Premix Ex Taq II (TaKaRa, Dalian, China) in accordance with the manufacturer’s instructions and were performed on an Icycler iQ5 (Bio-Rad, Berkeley, CA, USA). Three biological replications were performed for each sample, and three technical replications were performed for each biological sample. The actin gene was used as the internal reference gene. Relative expression levels were measured using the cycle threshold (Ct) 2^−ΔΔCt^ method. The primers for all the genes investigated are listed in Appendix A.

### 2.6. Transient Overexpression Experiments

For *Agrobacterium*-mediated transient expression, the full-length complete coding DNA sequences (CDS) of *PbERF22* (XM_009358931) were amplified and inserted individually into the vector pGreenII 62-SK to use for overexpression in ‘Zaosu’ fruit. The negative controls were infiltrated with *Agrobacterium* containing the pGreenII 62-SK-GUS and used for GUS staining. The GUS staining was performed as described by Zhai et al. [42]. All the constructs were transformed into *Agrobacterium* strain EHA105. The *Agrobacterium* EHA105 lines were incubated and resuspended to an OD_600_ of 0.6 in infiltration buffer containing MgCl_2_, MES, and acetosyringone. The fruit infiltration method used was as described by Zhai et al. [43]. At 3 d after injection, the peels around the injection sites were used for qRT-PCR. At 7 d after injection, the same sites were used for anthocyanin content and phenotype analyses. The primers for the constructed vectors are listed in Appendix A.

### 2.7. Dual Luciferase Assay

The promoter region of *PbUFGT* was amplified by PCR using specific primers. The PCR product was fused with the Firefly *luciferase (LUC)* reporter gene in the pGreen II 0800-LUC vector to obtain *PbUFGT* pro-LUC. The effector vectors of pGreen II 62-SK-PbERF22, pGreen II 62-SK-PbbHLH3, pGreen II 62-SK-PbMYB10, and -PbMYB10b were also constructed. All the constructed vectors were transformed into *Agrobacterium* strain GV3101 (psoup). They were transiently expressed in the agroinfiltrated leaves of *Nicotiana benthamiana*, and a subsequent dual LUC assay was performed. LUC and Renilla (REN) luciferase activities were assayed using the dual LUC assay kit (Promega, Madison, WI, USA), and the LUC/REN activity analysis was performed as described previously by An et al. [44]. Three assay measurements were performed for each construct. The primers for all the constructed vectors are listed in Appendix A.

### 2.8. Statistical Analyses

Statistical analyses of data were conducted using SPSS 23.0 software (SPSS, Chicago, IL, USA). Significant difference analyses were calculated using a one-way ANOVA test. The significant correlation analyses of gene transcription levels were performed using Pearson’s correlation analysis method based on *p* ≤ 0.05. Each value represents the mean ± standard deviation of three independent biological replicates.

## 3. Results

### 3.1. Changes in Coloration and Expression Levels of Anthocyanin Biosynthesis-Related Genes in Lanolin-Treated ‘Zaosu’ Pear

On the 10th day after receiving the lanolin treatment, the color of ‘Zaosu’ fruit peels were significantly redder than those of fruit growing under normal conditions (Figure 1a). The anthocyanin content after the lanolin treatment was significantly higher than that the control, which is consistent with the observed phenotype (Figure 1b). Moreover, the expression levels of the anthocyanin biosynthetic regulatory genes, *PbMYB10* and *PbMYB10b*, and the structural genes, *PbUFGT*, *PbANS*, and *PbDFR* significantly increased (Figure 1c). These results indicated that the lanolin treatment induced the red coloration of green-skinned ‘Zaosu’ pear fruit and promoted anthocyanin accumulation by up-regulating the expression levels of related structural and regulatory genes.

### 3.2. Transcriptome Data Analysis

To further investigate the lanolin-induced anthocyanin biosynthetic pathway, the untreated control and lanolin-treated ‘Zaosu’ fruit peels were subjected to an RNA-Seq analysis. To reduce the influence of the fruit development process and other factors, we selected two stages of lanolin-induced ‘Zaosu’ fruit coloration for the transcriptome analysis. This allowed us to quickly screen out the differentially expressed genes (DEGs) related to coloration. At 10 and 20 days after the lanolin treatment compared with the untreated control, 1351 and 2687 genes, having absolute values of log2 (fold change) ≥2, were identified as lanolin-induced DEGs, respectively. Among the DEGs, 748 were identified in both stages. Therefore, these 748 DEGs were selected as being potentially associated with the lanolin-induced coloration of ‘Zaosu’ fruit and were subjected to further functional analyses (Figure 2a). The KEGG enrichment analysis indicated that in addition to the metabolic pathways, other biological pathways produced extremely high P-values in this analysis. These indicated that the DEGs were significantly enriched in the biosynthesis of secondary metabolites, flavonoids, amino sugars, and nucleotide sugars, as well as plant hormone signal transduction and plant–pathogen interactions (Figure 2b). Moreover, a gene ontology (GO) annotation and enrichment analysis showed that the genes in the biological process group were mainly concentrated in the oxidation−reduction process, which is the molecular function term primarily concerned with oxidoreductase activity (Appendix A). These analyses indicated that multiple physiological mechanisms were involved in lanolin-induced anthocyanin biosynthesis. Here, the significant enrichment of DEGs in plant hormone signal transduction is of interest.

### 3.3. Lanolin-Induced Expression Levels of Jasmonate and Ethylene Synthesis-Related Genes

Based on the RNA-Seq analysis, we investigated the role of hormones in the lanolin-induced mechanism of anthocyanin biosynthesis. We found that the expression levels of the jasmonate synthetic genes *PbAOC*, *PbAOS*, *PbLOX*, and *PbOPR3* were significantly higher than those of the untreated control (Figure 3a). In addition, the expression levels of the ethylene synthetic genes *PbACO1* and *PbACS1* increased significantly in lanolin-induced pear compared with those in untreated controls (Figure 3b). Thus, the jasmonate and ethylene pathways are potentially involved in lanolin-induced anthocyanin biosynthesis.

### 3.4. Identification of PbERF22 As a Candidate Gene Involved in Anthocyanin Biosynthesis

Since both jasmonate and ethylene were involved in lanolin-induced responses, we focused on a class of *ERF* genes in the transcriptome data. *PbERF22*, a candidate gene, was screened from the RNA-Seq data. An qRT-PCR analysis showed that the expression level of *PbERF22* was significantly higher after the lanolin treatment compared with the control (Figure 4a), which was consistent with its transcript abundance in the transcriptome data. 

Furthermore, ‘Zaosu’ pear fruit were harvested at 100 DAFB, treated with 2 mmol/L MeJA, and stored at room temperature for 8 d. The *PbERF22* expression level increased significantly in MeJA-treated fruit compared with in controls (Figure 4b). Additionally, ‘Zaosu’ pear fruit at 90 DAFB were treated with 1-MCP, lanolin, or with 1-MCP followed by lanolin, and then, they were harvested 10 d later. The phenotype of pear fruit treated with lanolin was almost the same as those receiving the co-treatment of 1-MCP and lanolin (Figure 4c), which indicated that ethylene might not be a key factor, but only a participant, in lanolin-induced coloration. The expression levels of *PbACO1*, *PbACS1*, and *PbERF22* increased after the lanolin treatment. The expression levels of *PbACO1* and *PbACS1* increased significantly after the co-treatment of 1-MCP and lanolin compared with the 1-MCP treatment, and their expression levels decreased significantly compared with those in the lanolin treatment. This result demonstrated that 1-MCP successfully inhibited ethylene synthesis in the co-treated fruit. Compared with in the lanolin treatment, the expression level of *PbERF22* was significantly reduced in fruit co-treated with 1-MCP and lanolin (Figure 4d). These results indicated that the expression level of *PbERF22* was influenced by jasmonic acid and ethylene.

We analyzed the correlation between *PbERF22* and anthocyanin biosynthesis-related genes based on the transcriptional abundance levels of genes in the transcriptome data. *PbERF22* was significantly correlated with *PbMYB10* and *PbUFGT*, having correlation coefficients of 0.734 and 0.800, respectively (Table 1). In addition, the *PbERF22* expression levels during different developmental stages of several red-skinned pears were notably higher than those in ‘Zaosu’ pear (Appendix A). Therefore, we believed that *PbERF22* was related to lanolin-induced anthocyanin biosynthesis.

### 3.5. Transient Overexpression of PbERF22 in ‘Zaosu’ Pear

To verify *PbERF22*’s function in anthocyanin biosynthesis, the transient transformation of *PbERF22* was performed in young ‘Zaosu’ pear (Figure 5a; Appendix A). The overexpression of *PbERF22* resulted in the anthocyanin contents significantly increasing when compared with control fruit (Figure 5b). Compared with the empty vector, the significant up-regulation of *PbERF22* in fruit indicated that the transient transformation experiment was successful. The overexpression of *PbERF22* significantly up-regulated the expression levels of structural genes *PbCHS*, *PbDFR*, *PbANS,* and *PbUFGT* compared with their levels in controls (Figure 5c). Moreover, the expression levels of regulatory genes *PbMYB10*, *PbMYB10b*, and *PbbHLH3* were also up-regulated significantly (Figure 5d). Therefore, *PbERF22* promoted anthocyanin accumulation by up-regulating the expression levels of structural and regulatory genes.

### 3.6. Effects of PbERF22 on PbUFGT Promoter Activity

To verify how *PbERF22* regulates the expression levels of anthocyanin biosynthesis-related genes, we performed a dual LUC assay in tobacco leaves. *PbMYB10*, *PbMYB10b*, and *PbERF22* activated the promoter of *PbUFGT*. When *PbMYB10* or *PbMYB10b* was present, *PbERF22* enhanced the activation effects of *PbMYB10* and *PbMYB10b* on the *PbUFGT* promoter. Moreover, *PbERF22* enhanced the activity of the *PbUFGT* promoter in the presence of *PbMYB10* or *PbMYB10b* and *PbbHLH3*. Interestingly, compared with the co-transformation of *PbMYB10b* and *PbERF22*, the co-transformation of *PbERF22* and *PbMYB10* had a significantly greater activation effect on the *PbUFGT* promoter. Therefore, we believe that *PbERF22* significantly enhances the activation effect of the complex *MYB10*-*bHLH3* or *MYB10b*-*bHLH3* on the *PbUFGT* promoter.

## 4. Discussion

### 4.1. Lanolin Promotes Anthocyanin Biosynthesis in ‘Zaosu’ Pear Fruit

Initially, we noticed that several green-skinned pear fruit turned red when treated with lanolin during fruit development (Appendix A). Next, we studied the lanolin-treated ‘Zaosu’ fruit and found that lanolin induced red coloration on the sunny side of the pear fruit.

Many studies, using various methods to investigate anthocyanin synthesis in fruit, have been published. In strawberry, the use of red and yellow plastic films significantly increases the anthocyanin contents [45]. In pomegranate, chitosan coating treatments followed by cold storage reduce the anthocyanin degradation rate in juice and pomegranate arils [46]. In red Chinese sand pears, bagging, UV-B/visible irradiation, and jasmonic acid treatments affect the expression levels of genes related to anthocyanin synthesis and accumulation [47]. Thus, most red-skinned fruits have been used in anthocyanin biosynthetic studies, while the green-skinned fruits have been used less. In this study, although the lanolin application to induce anthocyanin biosynthesis was unique in pear fruit, the lanolin treatment resulted in the red coloration of green-skin pear and altered the fruits’ appearance. Lanolin promoted anthocyanin biosynthesis by inducing the up-regulation of genes related to anthocyanin biosynthesis (Figure 1). 

Many studies have shown that pure lanolin as negative controls, when applied to stems, leaves, and pedicels of plants [33,34,48], produces no significant physiological responses in these tissues because of its physical stability and chemical inertness. The cuttings of *Coleus* red were treated with diluted lanolin in greenhouses, its root formation is not affected [25]. However, a finding has shown that pure lanolin, when applied on chrysanthemum cuttings in summer, interferes with the rooting of cuttings [49]. These different effects of lanolin on the rooting of cuttings may be related to concentration of lanolin and environment of the treatment. In addition, lanolin affects pericarp development [36] and lateral bud germination [37]. In our study, anhydrous lanolin was applied to the fruit peel during fruit development, and this induced red coloration and physiological effects. Our preliminary study showed that lanolin induced red coloration because of the stimulation caused by lanolin itself rather than the temperature (Appendix A). Moreover, it caused physiological processes of flavonoid biosynthesis, plant hormone signal transduction, plant–pathogen interactions, and oxidation reduction (Figure 2; Appendix A). Therefore, we speculated that lanolin induced biological effects to plant tissues in some environments of treatment, and thus, it may not be an ideal negative control.

### 4.2. Lanolin Affects the Levels of Several Hormones in ‘Zaosu’ Pear Fruit

Plant hormones are important signal transducers in plant cells that respond to the external environment [50,51] and vital factors affecting anthocyanin biosynthesis [52,53,54]. The RNA-Seq analysis showed that DEGs were significantly enriched in plant hormone signal transduction (Figure 2b). Therefore, we measured the contents of several hormones in lanolin-treated ‘Zaosu’ pear fruit and found that the jasmonate contents were significantly increased and the abscisic acid contents were significantly decreased (Appendix A). Moreover, through a transcription factor family analysis, we found that ERFs were highly enriched (Appendix A), which caused us to speculate that the ethylene synthesis pathway had proceeded in lanolin-treated ‘Zaosu’ pear fruit. 

A gene expression analysis further indicated that the jasmonate synthetic genes *PbAOC*, *PbAOS*, *PbLOX*, and *PbOPR3* and the ethylene synthetic genes *PbACS1* and *PbACO1* were significantly up-regulated (Figure 3). *PbAOC*, *PbAOS*, *PbLOX*, and *PbOPR3* are the key genes of the jasmonate synthetic pathway [55], while *PbACS1* and *PbACO1* are the key genes of the ethylene synthetic pathway [56]. Although less ethylene production occurred in the 60 DAFB period of ‘Zaosu’ fruit, which might have led us to fail to measure ethylene production, the gene expression (Figure 3) and the transcription factor family (Appendix A) analyses indicated that ethylene production was also induced by lanolin. Therefore, we speculated that the crosstalk among multiple hormones was involved in lanolin-induced anthocyanin biosynthesis.

### 4.3. PbERF22 Responses to Jasmonate and Ethylene and is Correlated with Anthocyanin Biosynthesis

Since lanolin treatments resulted in anthocyanin accumulation and jasmonate and ethylene changes, we focused on a class of *ERF* genes found in the transcriptome data. *ERF* genes respond to hormone signals, including those of jasmonate and ethylene [57]; ERFs are important components downstream in the ethylene signaling and response pathway, and they are involved in various ethylene-mediated responses [58,59]. *ERF* genes, such as *AtERF2* [60] and *BrERF72* [61], also positively respond to jasmonate signal-transduction pathways, and their expression levels are regulated synergistically by both ethylene and jasmonate. *ERF1* expression can be synergistically activated by jasmonate and ethylene to regulate defense–response genes [17]. In soybean, the jasmonic acid and ethylene treatments induce the expression of *GmERF3*, which has an important role in responses to biotic and abiotic stresses [62]. In tobacco, *JERF1* acts as a transcription factor in different signal-transduction pathways. It binds to the GCC box or dehydration-responsive element(DRE) sequence and responds to jasmonic acid, ethylene, and abscisic acid signals [63]. In this study, *PbERF22* was selected as a candidate gene based on an RNA-Seq data analysis. A qRT-PCR analysis showed that the *PbERF22* expression level was markedly up-regulated by lanolin (Figure 4a). Multiple cis-acting MeJA-responsive elements, including CGTCA and TGACG motifs [64], were identified by analyzing the *PbERF22* promoter sequence. Furthermore, we observed that the MeJA treatment promoted the *PbERF22* expression and that 1-MCP significantly reduced the *PbERF22* expression induced by lanolin. The expression patterns of *PbACS1* and *PbERF22* were highly similar, suggesting that *PbERF22* responds to jasmonate and ethylene (Figure 4). 

In addition, *ERF* genes participate in the regulation of anthocyanin biosynthesis. The *AtERF4* and *AtERF8* double-mutant reduces the light-induced anthocyanin contents in Arabidopsis [65]. Under low-isoelectric point conditions, the anthocyanin contents in Arabidopsis lines that overexpress *JcERF035* decrease compared with the controls [21]. Here, a phylogenetic analysis found that *PbERF22* and *PyERF3* were clustered together in one clade (Appendix A). *PyERF3* is involved in the regulation of anthocyanin biosynthesis in pears [22]. Furthermore, *PbERF22* was highly correlated with the expression levels of *PbMYB10* and *PbUFGT* (Table 1), and the *PbERF22* expression levels in several red-skinned pears were notably higher than those in ‘Zaosu’ pears (Appendix A). Therefore, we speculated that *PbERF22* was involved in anthocyanin biosynthesis and was an important link between the jasmonate and/or ethylene hormone signals and anthocyanin biosynthesis.

### 4.4. Functional Validation of PbERF22 in Anthocyanin Biosynthesis

To test the hypothesis that *PbERF22* is involved in anthocyanin biosynthesis, we investigated *PbERF22*’s function in anthocyanin biosynthesis. The overexpression of *PbERF22* promoted anthocyanin biosynthesis by up-regulating the expression levels of anthocyanin structural and regulatory genes in ‘Zaosu’ pear fruit (Figure 5). *UFGT*, a key gene in anthocyanin biosynthesis, is regulated by the MBW complex [7]. Transcription factors, including ERFs, are involved in the regulation of anthocyanin synthesis through interactions with the MBW complex. ERF3 forms complexes with MYB10 and bHLH3 or MYB114 plus bHLH3 to enhance binding to the *UFGT* promoter and the co-regulation of anthocyanin biosynthesis [22]. *Pp4ERF24* and *Pp12ERF96* facilitate the *PpUFGT* activity level by enhancing interactions with *PpMYB114* and *PpbHLH3* [23]. *MYB10* and *MYB10b* positively regulate the expression levels of structural genes in the anthocyanin biosynthetic pathway in pear [43,66,67]. Therefore, in this study, we investigated whether *PbERF22* has effects on the ability of *PbMYB10* and *PbMYB10b* to activate the promoter of *PbUFGT*. We determined that *PbERF22* enhances the activation effect of *PbMYB10* or *PbMYB10b* on the *PbUFGT* promoter. This result was similar to that of *PyERF3*, which enhances the transcriptional activities of *PyMYB10* and *PyMYB114* [22]. However, *PbERF22* showed more significant effects on the transactivation of *PbMYB10* or the complex of *PbMYB10* plus *PbbHLH3* on the *PbUFGT* promoter (Figure 6). 

## 5. Conclusions

We found that lanolin can induce the red appearance of green-skinned pears, and that multiple pathways were involved in the complex coloring mechanism. *PbERF22* was confirmed to respond to jasmonate and ethylene signals and to promote anthocyanin biosynthesis by enhancing the activation effects of *MYB10* and *MYB10b*. We believe that *PbERF22* plays an important role in the interactions between hormones and anthocyanin biosynthesis. The lanolin-induced red phenomenon will allow us to gain insights into more mechanisms that influence the anthocyanin synthetic pathway in green-skinned pears.

## Figures and Tables

**Figure 1 biomolecules-10-00278-f001:**
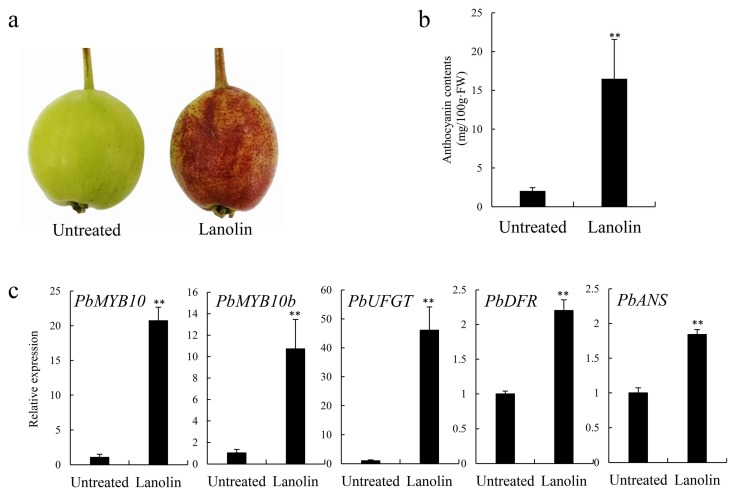
Anthocyanin biosynthesis was induced by a lanolin treatment in ‘Zaosu’ pear fruit at 10 d after treatment. (**a**) For untreated control and lanolin-treated ‘Zaosu’ fruit phenotypes of peels at 10 days after treatment; (**b**) total anthocyanin contents; and (**c**) expression levels of anthocyanin biosynthetic structural and regulatory genes were determined. ** indicates significant differences at *p* < 0.01. All the data are means ± SDs of three biological replicates.

**Figure 2 biomolecules-10-00278-f002:**
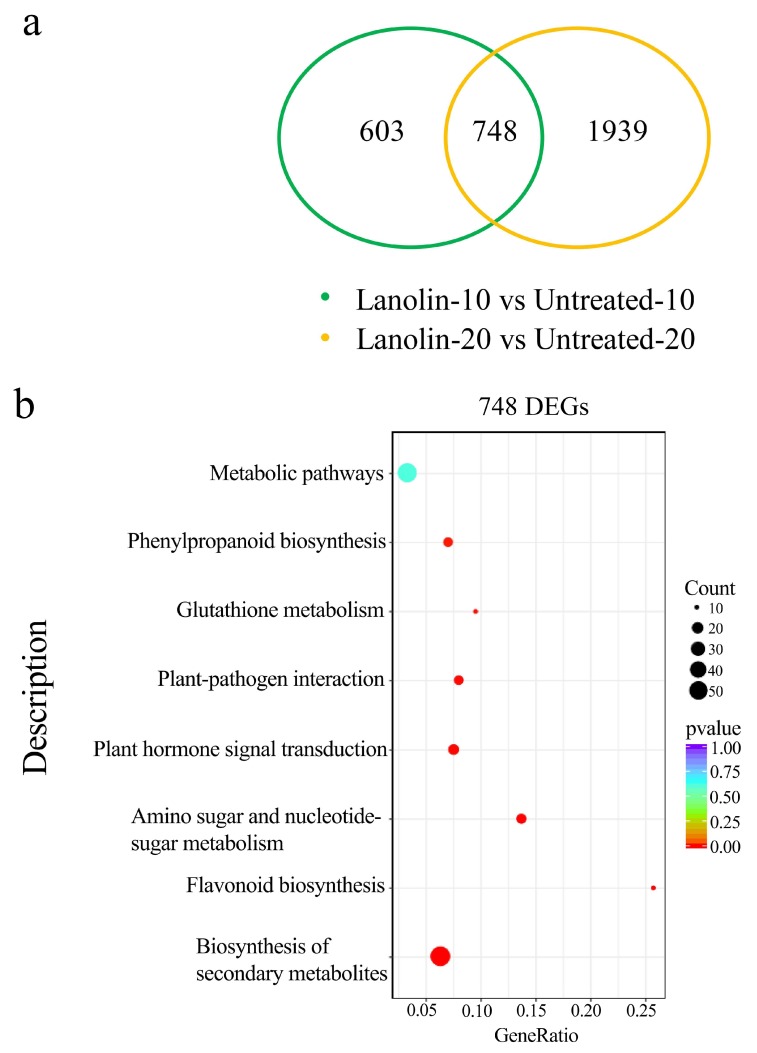
RNA-Seq analysis of untreated control and lanolin-treated ‘Zaosu’ pear fruit peels. (**a**) Venn diagram of the numbers of differential expression genes (DEGs) between untreated and lanolin-treated fruit peels at 10 and 20 days after treatment. (**b**) The Kyoto Encyclopedia of Genes and Genomes (KEGG) enrichment analysis of 748 DEGs.

**Figure 3 biomolecules-10-00278-f003:**
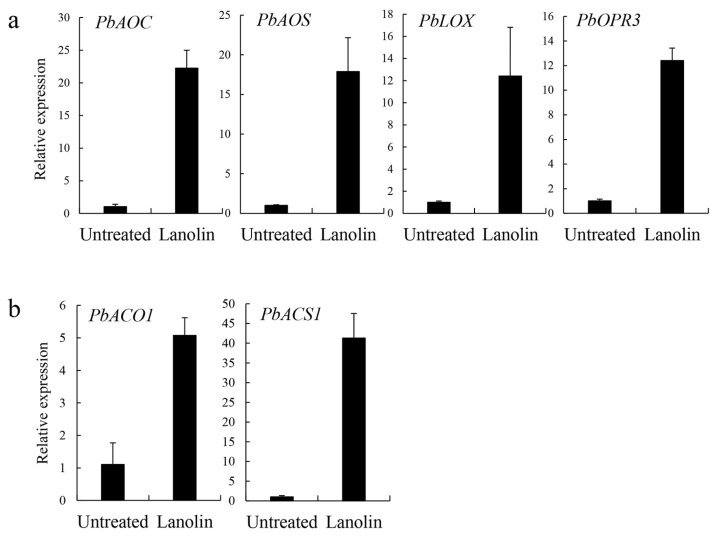
The expression analysis of jasmonate- and ethylene synthesis-related genes. The expression levels of (**a**) jasmonate and (**b**) ethylene synthetic genes in lanolin-treated and untreated control ‘Zaosu’ pear fruit after 10 days of treatment. Data are means ± SDs of three biological replicates.

**Figure 4 biomolecules-10-00278-f004:**
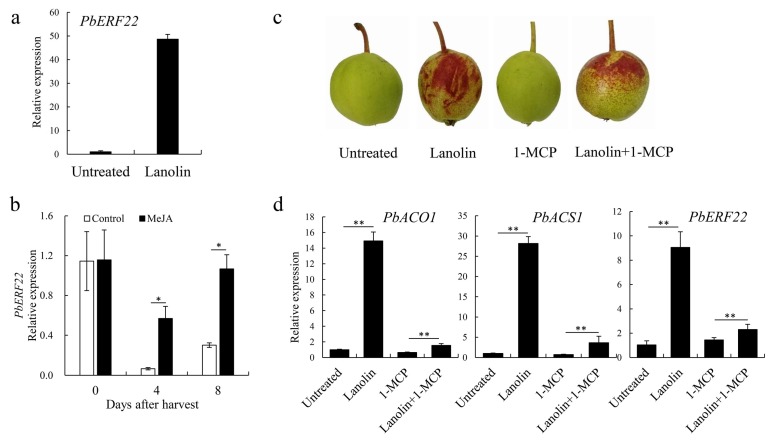
*PbERF22* responded to jasmonate and ethylene and was selected as a candidate gene related to anthocyanin biosynthesis. (**a**) Quantitative real-time PCR analysis of *PbERF22* expression levels in lanolin-treated and untreated control ‘Zaosu’ fruit; (**b**) *PbERF22* expression levels after methyl jasmonate (MeJA) and water (control) treatments. ‘Zaosu’ pear fruit were harvested at 100 days after full bloom (DAFB), treated independently with 2 mmol/L MeJA and water, and then stored at room temperature for 8 d; (**c**) The phenotypes of ‘Zaosu’ pear and treated with lanolin, 1-methylcyclopropene (1-MCP; an ethylene antagonist), or with lanolin followed by 1-MCP. Untreated pear served as the controls; (**d**) The expression levels of *PbACO1*, *PbACS1*, and *PbERF22* in the above treated pear fruit. In (**c**) and (**d**), fruit on the ‘Zaosu’ fruit tree were treated at 90 DAFB with 1-MCP or lanolin, or with 1-MCP followed by lanolin and then obtained 10 days after treatment. * indicates significant differences at *p* < 0.05, ** indicates significant differences at *p* < 0.01. Error bars for the data represent ± SDs of three biological replicates.

**Figure 5 biomolecules-10-00278-f005:**
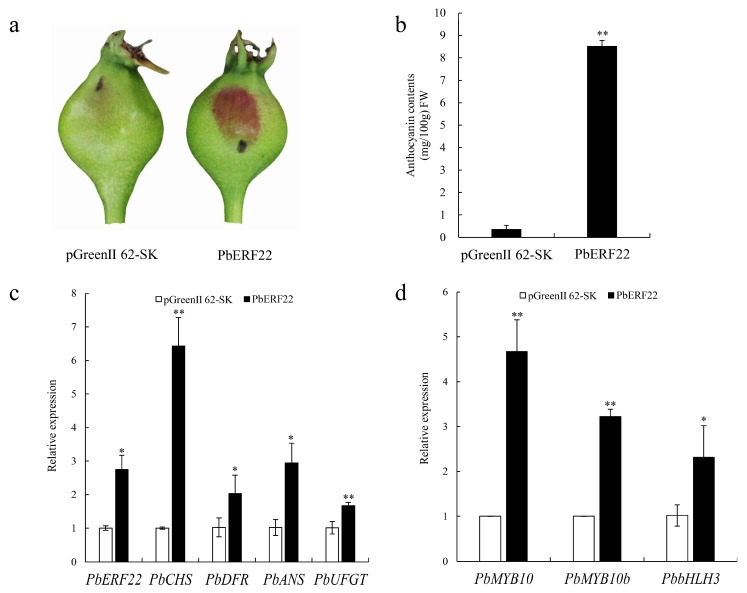
Transient transformation assays verifying *PbERF22*’s function in the coloration of ‘Zaosu’ fruit peel. (**a**) The phenotype of the ‘Zaosu’ pear 7 d after overexpressing *PbERF22* and the empty vector (pGreenII 62-SK), which were introduced by infiltration; (**b**) Measurements of anthocyanin contents in the injection regions of fruit peel; (**c**) qRT-PCR analysis of the expression levels of anthocyanin structural genes after infiltration; (**d**) The expression levels of anthocyanin regulatory genes. * indicates significant differences at *p* < 0.05, ** indicates significant differences at *p* < 0.01. Error bars represent ± SDs of three biological replicates.

**Figure 6 biomolecules-10-00278-f006:**
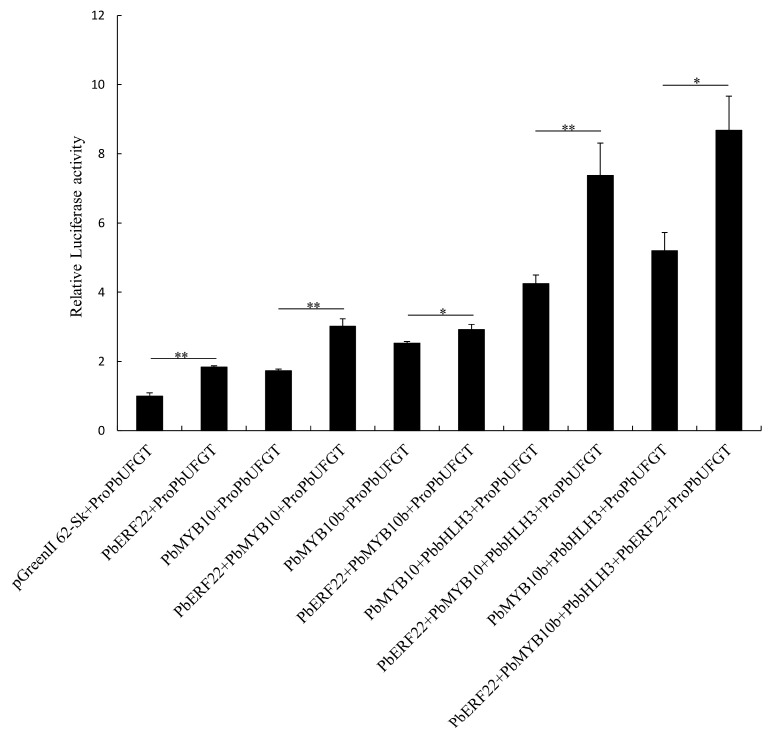
Dual luciferase (LUC) assay verifying that *PbMYB10*, *PbMYB10b*, and *PbbHLH3* co-transformed with *PbERF22* increased the *PbUFGT* promoter activity. The *PbUFGT* promoter activity is expressed as a ratio of LUC to Renilla (REN) activities. * indicates significant differences at *p* < 0.05, ** indicates significant differences at *p* < 0.01. Data are means ± SEs of three biological replicates.

**Table 1 biomolecules-10-00278-t001:** Correlation analysis between *PbERF22* and anthocyanin biosynthesis-related genes.

Correlation of Factor	*PbERF22*	*PbMYB10*	*PbMYB10b*	*PbUFGT*	*PbDFR*	*PbANS*
*PbERF22*	1	0.734 ^**^	0.565	0.800 ^**^	0.456	0.121
*PbMYB10*		1	0.469	0.749 ^**^	0.399	0.581 ^*^
*PbMYB10b*			1	0.769 ^**^	0.693 ^*^	−0.159
*PbUFGT*				1	0.689 ^*^	−0.066
*PbDFR*					1	−0.114
*PbANS*						1

*: Correlation is significant at the 0.05 level (*p* < 0.05, two tailed); **: Correlation is significant at the 0.01 level (*p* < 0.01, two tailed).

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
