# Peer review of "Jasmonate and Ethylene-Regulated Ethylene Response Factor 22 Promotes Lanolin-Induced Anthocyanin Biosynthesis in ‘Zaosu’ Pear (Pyrus bretschneideri Rehd.) Fruit"

_biomolecules, 2020, doi:10.3390/biom10020278_

Round 1
Reviewer 1 Report
This research Is interesting AND with good Scientific work. I suggest un the discussion section explain some possible physiological mechanisms by which probable lanolin induce anthocyanins. The rest Is nice presented. When corrected this suggestion, I suggest to accept this manuscript for publicación.
Author Response
Dear editor and reviewers:
Thanks for your comments and suggestions. Your comments were highly insightful and enabled us to greatly improve the quality of our manuscript. We have revised our manuscript (ID: biomolecules-699886) according to the comments. In the following pages are our point-by-point responses to each of the comments of the reviewers. To help reviewers track easily the revisions we made, the revised portions are marked in red in the paper, and strikethrough font [example] for deletions.
We appreciate for Editors/Reviewers’ warm work earnestly and hope that the correction will meet with approval. Once again, thank you very much for your comments and suggestions.
Yours sincerely,
Lingfei Xu
Add:College of Horticulture Northwest A&F University
Yangling, Shaanxi Province, China, 712100
Telephone:86-29-87082150;Mobile:13700284885
Fax:86-29-87082149
E-mail:lingfxu2013@sina.com
Point: This research Is interesting AND with good Scientific work. I suggest on the discussion section explain some possible physiological mechanisms by which probable lanolin induce anthocyanins. The rest is nice presented. When corrected this suggestion, I suggest to accept this manuscript for publicación.
Response: We are encouraged by this reviewer’s comment, we have explained some possible physiological mechanisms by which probable lanolin induce anthocyanins on the discussion section. We have added the discussion information in line 337-338 and 340-342.
Reviewer 2 Report
Dear Editor, in the present manuscript (Biomolecules-699886) authors investigated the anthocyanin biosynthesis process on lanolin-treated ‘Zaosu’ pear (Pyrus bretschneideri Rehd.) fruit by using a comparative transcriptome analysis. In general, the experiments were well developed and the manuscript well written showing new and interesting information regarding the pivotal role played by the ethylene response factor PbERF22 on this process since its expression responses to jasmonate and ethylene signals and regulates anthocyanin biosynthesis throughout regulation of anthocyanin biosynthetic structural and regulatory genes. Thus, it could be suitable for publication in Biomolecules after minor revision.
- Line 51: Consider adding reference to support this statement.
- Lines 84 and 93: Were these treatments performed in fruits on the same tree or in different tress for each replicate?
- Line 84: If 20 fruits were used for each treatment, and 3 replicates of 5 fruits were taken after 10 and 20 d, no enough number of fruits is available. Check number of fruit.
- Line 100: Why fruits from a different location were used in this experiment?
- Line 232-234 and 297-299 contain information more appropriate for Discussion section. Authors might consider writing results and discussion in a single section, which probably would make easier to read and understand the observed findings.
- Line 193: Consider adding “at 10 d after treatment” after determined.
- Line 230: Add “after x days of treatment “ after fruit.
- Lines 267-268: Unclear sentence. Please re-write as “In (c) and (d) ‘Zaosu’ pear fruit were harvested at 90 DAFB, treated with 1-MCP or lanolin, or with 1-MCP followed by lanolin treatment and then stored at ----- temperature for 10 d.”
- Write paper tittle on reference list according to the journal format.
